REGISTERED REPORT

# Registered report: Senescence surveillance of pre-malignant hepatocytes limits liver cancer development

**Samrrah Raouf[1], Claire Weston[2], Nora Yucel[3], Reproducibility Project: Cancer Biology\*[†]**

[1]Mouse Biology Program, University of California–Davis, Davis, California; [2]Reveal Biosciences, San Diego, California; [3]Stanford University, Stanford, California

**Abstract** The Reproducibility Project: Cancer Biology seeks to address growing concerns about reproducibility in scientific research by conducting replications of 50 papers in the field of cancer biology published between 2010 and 2012. This Registered report describes the proposed replication plan of key experiments from 'Senescence surveillance of pre-malignant hepatocytes limits liver cancer development' by *Kang et al. (2011)*, published in Nature in 2011. The experiments that will be replicated are those reported in Figures 3B, 3C, 3E, and 4A. In these experiments, *Kang et al. (2011)* demonstrate the phenomenon of oncogene-induced cellular senescence and immune-mediated clearance of senescent cells after intrahepatic injection of *NRAS* (Figures 2I, 3B, 3C, and 3E). Additionally, *Kang et al. (2011)* show the specific necessity of CD4+ T cells for immunoclearance of senescent cells (Figure 4A). The Reproducibility Project: Cancer Biology is a collaboration between the Center for Open Science and Science Exchange, and the results of the replications will be published by *eLife*.

**\*For correspondence:** joelle@ scienceexchange.com

**Group author details**
[†]Reproducibility Project: Cancer Biology
See page 13

**Reviewing editor**: Ronald N Germain, National Institute of Allergy and Infectious Diseases, United States

## Introduction

Cellular senescence—a permanent state of proliferative arrest—has been shown to be an important failsafe mechanism against tumor development in vivo. Aberrant activation of oncogenes can force normal cells into a senescent state of stable cell cycle arrest (*Narita and Lowe, 2005*). Senescence is often associated with a pre-malignant state, and a wide range of cancers have been shown to harbor populations of senescent cells (*Quintanilla et al., 1986*; *Braig et al., 2005*; *Collado et al., 2005*; *Bennecke et al., 2010*). An ongoing debate in the field of cancer biology has been the phenomenon of 'immune surveillance', or the ability of the immune system to specifically identify and eliminate nascent precancerous (e.g., senescent) cells before they can cause harm (*Swann and Smyth, 2007*). The possibility of tailored, pro-senescent therapies for cancer treatment has provoked considerable interest.

*Kang et al. (2011)* report that pre-malignant, senescent hepatocytes are recognized and cleared through a CD4+ T-cell-specific immune response and that this immunosurveillance is crucial for tumor suppression in vivo. The authors attempted to mimic aberrant oncogene activation by stably delivering oncogenic $Nras^{G12V}$ into mouse livers in vivo. Hydrodynamic injection of transposable elements resulted in mosaic generation of *Nras*-expressing hepatocytes that displayed senescent markers. Time course analyses revealed a progressive loss of $Nras^{G12V}$ expressing senescent cells within 2 months after stable intrahepatic delivery of oncogenic $Nras^{G12V}$. Studies were carried out in wild-type mice, as well as immunocompromised mice (SCID/beige and $CD4^{-/-}$). Importantly, all analyses were paralleled using an $Nras^{G12V}$ effector loop mutant ($Nras^{G12V/D38A}$) incapable of signaling to downstream pathways (*Kang et al., 2011*).

In Figures 2I, 3B, 3C, and 3E, *Kang et al. (2011)* tested whether intrahepatic expression of oncogenic $Nras^{G12V}$ induced cellular senescence in affected hepatocytes (as indicated by the expression of

pro-senescent markers p16 and p21), and if senescent cells were eventually cleared over a time course spanning 60 days. Further, Kang et al. demonstrated the necessity of a functional adaptive immune response in clearing senescent cells, as mice harboring mutations in adaptive immunity (SCID/beige) showed significant attenuation in the ability to clear pre-malignant cells. These key experiments, which test the major underlying hypotheses of the paper, are replicated with Protocol 1.

In Figure 4A, Kang et al. demonstrated the specific necessity of CD4$^+$ T lymphocytes in undertaking senescence surveillance. They showed that CD4$^{-/-}$ mice displayed significantly abrogated clearing of Nras-positive hepatocytes 12 days following intrahepatic injection of NrasG12V; no immunoclearance was observed for either CD4$^{-/-}$ mice or wild-type controls expressing the defective signaling mutant NrasG12V/D38A. These key results suggest that an intact CD4$^+$ T-cell-mediated adaptive immune response is necessary for senescence surveillance of pre-malignant hepatocytes. These experiments will be replicated in Protocol 2.

To date, conflicting reports of Ras-associated senescence have been reported. Using a mouse model with conditional pancreatic expression of *Kras*, *Kennedy et al. (2011)* demonstrated that activated KrasG12D-induced senescence in pancreatic cells. However, using an intrahepatic injection system similar to Kang et al., *Ho et al. (2012)* did not detect positive markers for senescence in the livers of *Nras*-injected wild-type mice as compared to controls. Multiple studies have observed immunoclearance of senescent cells similar to results reported by Kang et al. *Xue et al. (2007)* reported rapid clearance of senescent cells by the immune system, and *Rakhra et al. (2010)* reported similar findings, as well as demonstrating that CD4$^+$ T cells were necessary for clearing senescent cells. Similarly, *Acosta et al. (2013)* demonstrated clearance of Nras-positive cells in wild-type mice, but impaired clearance in mice treated with various immunosuppressive drugs. Interestingly, co-injection of NrasG12V along with other oncogenes failed to manifest senescence surveillance, and instead resulted in an aggressive tumor phenotype (*Brinkhoff et al., 2014*). Likewise, intrahepatic activation of SV40 large T antigen resulted in hepatocellular carcinoma development and did not activate immunosurveillance (*Willimsky and Blankenstein, 2005*).

## Materials and methods

Unless otherwise noted, all protocol information was derived from the original paper, references from the original paper, or information obtained directly from the authors. An asterix (*) indicates data or information provided by the Reproducibility Project: Cancer Biology core team. A hashtag (#) indicates information provided by the replicating lab.

### Protocol 1: generation of oncogene-induced senescence and immunosurveillance in murine hepatocytes

This protocol assesses whether intrahepatic expression of oncogenic *Nras$^{G12V}$* induces cellular senescence, and if such pre-malignant senescent hepatocytes are eventually eradicated over time, as is depicted in Figures 2I, 3B, 3C, and 3E. This protocol also determines the necessity of a functional adaptive immune response in clearing senescent cells by measuring the abrogation of Nras-, p21-, and p16-positive hepatocytes in severely immune-compromised mice (SCID/beige) injected with either *Nras$^{G12V}$* or *Nras$^{G12V/D38A}$*.

### Sampling

- These experiments will analyze a minimum of five mice per treatment group, for a total power of between 89.8% and 99.9%.

  1. See 'Power calculations' section for details.
  2. In order to account for variability in hydrodynamic injections that might contribute to exclusion of study animals (estimated as a 20% failure rate by Kang et al.), the initial starting sample size will be seven mice per treatment group. The inclusion of two additional animals will help ensure that at least five animals are available for analysis at the conclusion of the study.

- Outline of experimental conditions:

  1. Mouse Cohort 1 (tissue processed at 6 days post-injection).

     - 14 female C.B-17 wild-type mice (4–6 weeks old).

       a. Seven mice injected with pPGK-SB13 + pT/CaggsNras$^{G12V}$
       b. Seven mice injected with pPGK-SB13 + pT/CaggsNras$^{G12V/D38A}$

- 14 female C.B-17 SCID/beige mice (4–6 weeks old).

  a. Seven mice injected with pPGK-SB13 + pT/CaggsNras$^{G12V}$
  b. Seven mice injected with pPGK-SB13 + pT/CaggsNras$^{G12V/D38A}$

2. Mouse Cohort 2 (tissue processed at 30 days post-injection).

- 14 female C.B-17 wild-type mice (4–6 weeks old).

  a. Seven mice injected with pPGK-SB13 + pT/CaggsNras$^{G12V}$
  b. Seven mice injected with pPGK-SB13 + pT/CaggsNras$^{G12V/D38A}$

- 14 female C.B-17 SCID/beige mice (4–6 weeks old).

  a. Seven mice injected with pPGK-SB13 + pT/CaggsNras$^{G12V}$
  b. Seven mice injected with pPGK-SB13 + pT/CaggsNras$^{G12V/D38A}$

## Materials and reagents

| Reagent | Type | Manufacturer | Catalog # | Comments |
|---------|------|--------------|-----------|----------|
| pPGK-SB13 transposase vector | DNA Vector | N/A | N/A | Original reagent obtained from authors |
| pT/CaggsNrasG12V transposon vector | DNA Vector | N/A | N/A | Original reagent obtained from authors |
| pT/CaggsNrasG12V/D38A transposon vector | DNA Vector | N/A | N/A | Original reagent obtained from authors |
| GenElute Endotoxin-free Plasmid Maxiprep Kit | Reagent | Sigma | PLEX15-1KT | This kit replaces the Qiagen Endo-free Maxiprep kit used by the original authors |
| Ketamine HCl | Anesthetic | | | Specific brand information will be left up to the discretion of the provider lab and recorded later |
| Acepromazine maleate | Anesthetic | | | |
| Butorphanol tartrate | Anesthetic | | | |
| Dulbecco's Phosphate Buffered Saline | Reagent | Sigma–Aldrich | D1408 | Original brand not specified |
| Paraformaldehyde | Reagent | Sigma–Aldrich | 158127 | Original brand not specified |
| Mouse anti-Nras (F155) | Antibody | Santa Cruz | Sc-31 | Same as original |
| Mouse anti-p21 (SXM30) | Antibody | BD Pharmingen | 556431 | Same as original |
| Rabbit anti-p16 (M156) | Antibody | Santa Cruz | sc-1207 | This clone is no longer available. We will work with the provider lab to determine a suitable replacement |
| Biotinylated goat anti-mouse IgG$_1$-conjugate | Antibody | Thermo-Fisher | TM-060-BN | Same as original |
| Biotinylated goat anti-rabbit IgG- conjugate | Antibody | Thermo-Fisher | TR-060-BN | Same as original |
| IgG1 mouse isotype control antibody | Antibody | Sigma–Aldrich | M5284 | Originally not included |
| IgG rabbit isotype control antibody | Antibody | | | We will obtain a suitable rabbit IgG control antibody if necessary |
| Streptavidin-HRP conjugate | Reagent | Sigma–Aldrich | GERPN1231 | Original not specified |

*Table 1. Continued on next page*

*Table 1. Continued*

| Reagent | Type | Manufacturer | Catalog # | Comments |
|---|---|---|---|---|
| Diaminobenzidine (DAB) | IHC Stain | | | Specific brand information will be left up to the discretion of the provider lab and recorded later |
| Haematoxylin | IHC Stain | | | |
| Eosin | IHC Stain | | | |
| 4-week-old female Fox Chase CB17 (C.BKa-*Igh*$^b$/IcrCrl) | Mouse | Charles River | Strain 251 | |
| 4-week-old female Fox Chase SCID Beige (CB17.Cg-*Prkdc*$^{scid}$*Lyst*$^{bg-J}$/Crl) | Mouse line | Charles River | Strain 250 | |

## Procedure

Note: The following procedures (steps 1–5) are presented in greater detail in *Bell et al. (2007)*. Please consult this resource for enhanced experimental protocol detail and description.

1. Grow and prepare endotoxin-free plasmid constructs according to the manufacturer's protocol for the GenElute Endotoxin-free Plasmid Maxiprep Kit. Store DNA at ~1–2 µg/µl in 10 mM Tris–HCl, pH 7.2, 0.1 mM ethylenediaminetetraacetic acid (EDTA) at 4°C.

   a. Prepare >400 µg of pPGK-SB13 transposase vector.
   b. Prepare >1000 µg of pT/CaggsNras$^{G12V}$ transposon vector.
   c. Prepare >1000 µg of pT/CaggsNras$^{G12V/D38A}$ transposon vector.

2. Sequence plasmids to confirm identity and run on gel to confirm vector integrity, as well as *Nras* mutational status.

   a. Use the following pT/CaggsNras sequencing primers:

      i. Forward: tgtgaccggcggctctaga.
      ii. Reverse: cgaggctgatccttgaaagtggctctt.

3. Obtain 4-week-old female C.B-17 wild-type and C.B-17 SCID/beige mice.

   a. Mice should be randomized into strain-based treatment and control groups and co-housed so that mice of the same strain receiving either *Nras*$^{G12V}$ or *Nras*$^{G12V/D38A}$ injections are housed together.
   b. House mice in ventilated cages (IVC) that are specific pathogen free (SPF). Allow mice 1 week to acclimate to new housing prior to injection.

4. Prepare 5-week-old mice for injection.

   a. Mix anesthetic cocktail: 8 mg/ml ketamine HCl, 0.1 mg/ml acepromazine maleate, and 0.01 mg/ml butorphanol tartrate.
   b. Weigh mice.
   c. Administer anesthetic to mice; avoid rendering them unconscious (mice should appear 'drowsy').

      i. For mice 25–30 g, inject 50 µl of cocktail *i.p.*
      ii. For mice <20 g, inject 25–30 µl of cocktail *i.p.*

5. Inject 5-week-old mice with transposon/transposase:

   a. Mix at a 5:1 molar ratio of transposon to transposase-encoding plasmid for a total injection mass of 30 µg.

      i. 25 µg pT/CaggsNras$^{G12V}$ or pT/CaggsNras$^{G12V/D38A}$.
      ii. 5 µg pPGK-SB13.

   b. Weigh mice.

 c. Suspend 30 µg of plasmids in 0.9% NaCl at a final volume of 10% of the animal's body weight (upper limit is 2.5 ml injection volume).

 d. Blindly deliver DNA solution by hydrodynamic tail vein injection within a period of 4–7 s.

 i. Exclusion criteria for injection: as soon as resistance is encountered during tail-vein injection, the respective animal cannot be included for studies. Injections taking >10 s are considered unsuccessful and are not included in test groups.

6. At days 6 and 30 post-injection of DNA, euthanize mice and harvest liver tissue.

 a. Tissues should be randomized and blinded and shipped to Reveal Bioscience for downstream analysis.

7. Fix tissue and embed in paraffin using the following procedure:

 a. Fix tissues with 4% paraformaldehyde (PFA) overnight at 4°C.

 b. Rinse tissues with phosphate-buffered saline (PBS) (five washes for 1 hr each at 4°C).

 c. Dehydrate tissues using a step-wise gradient from 70–100% ETOH and xylene using an automatic tissue processor.

 d. Embed tissues in paraffin.

8. Section tissues and perform immunohistochemistry using the following procedure:

 a. Use a microtome to prepare 2–3 µm sections.

 b. Mount sections on slides.

 c. Deparaffinize sections using xylene and ETOH and rehydrate.

 d. Perform heat-induced epitope retrieval using sodium citrate, pH 6.0.

 e. Rinse sections with tris-buffered saline (TBS) (one wash for 3 min).

 f. Block tissues with 3% hydrogen peroxide ($H_2O_2$) for 10 min.

 g. Rinse sections with TBS (one wash for 3 min).

 h. Block sections with blocking solution for 5 min.

 i. Rinse sections with TBS (one wash for 3 min).

 j. Incubate sections with primary antibody overnight at 4°C.

 k. Rinse sections with TBS (one wash for 3 min).

 l. Incubate sections with secondary antibody for 30 min at RT.

 m. Rinse sections with TBS (one wash for 3 min).

 n. Incubate sections with Streptavidin-HRP for 10–15 min at RT.

 o. Rinse sections with TBS (one wash for 3 min).

 p. Incubate sections with 3,3' Diaminobenzidine (DAB) + DAB substrate.

 q. Rinse sections with $ddH_2O$.

 r. Counterstain sections with Haematoxylin for 30 s.

 s. Rinse sections with $ddH_2O$ for 10–15 min.

 t. Dehydrate and mount sections for imaging.

9. Perform immunohistochemistry staining on each tissue section with the following:

 a. Primary antibodies for N-ras, p21, and p16.

 i. mouse-anti-Nras, 1:100 dilution.

 ii. mouse-anti-p21, 1:50 dilution.

 iii. rabbit-anti-p16, 1:50 dilution.

 b. Secondary antibodies.

 i. Biotinylated goat anti-mouse $IgG_1$-conjugate.

 ii. Biotinylated goat anti-rabbit IgG- conjugate.

 c. Isotype control primary antibody + secondary antibody.

 i. IgG1 mouse isotype control antibody.

 ii. IgG rabbit isotype control antibody.

 d. Secondary antibody only.

10. Blindly examine five 200× fields from two stained liver sections from each mouse.

 a. Count 200 total cells per field and record number of Nras-, p21-, and p16-positvely stained cells for each image.

## Deliverables

- Data to be collected:

1. Sequencing information and gel-verification of pT/CaggsNras plasmids.
2. Mouse health records (weight at time of injection, duration(s) of injection, health over course of experiment, etc).
3. Images of all stained sections, including controls. (Compare visually to Figure 2I).
4. Raw total cell counts and Nras-, p21-, and p16-positively stained cells for each field examined.
5. Bar graph of Nras-, p21-, and p16-positively stained cells as a percentage of total cells in field for each condition. (Compare to Figures 3B, 3C, and 3E).

- Samples delivered for further analysis:

1. Purified transposon/transposase plasmids will be used in Protocol 2.

## Confirmatory analysis plan

This replication attempt will perform the following statistical analyses listed below. First, the replication effect size will be computed and compared to the original effect size for each analysis. Second, the original effect size and the replication effect size will be combined into a single effect size using a meta-analytic approach and will be presented as a forest plot.

- Statistical Analysis of the Replication Data:

1. Three-way ANOVA (2 × 2 × 2), comparing the percent of Nras-positive cells in C.B-17 wild-type and C.B-17 SCID/beige mouse livers 6 and 30 days after stable delivery of $Nras^{G12V}$ or $Nras^{G12V/D38A}$.

 - Two-way ANOVA comparing the percent of Nras-positive cells in C.B-17 wildtype and C.B-17 SCID/beige mouse livers 6 and 30 days after stable delivery of $Nras^{G12V}$.

 a. Planned comparisons with the Bonferroni correction:

 i. C.B-17 wild-type mouse livers 6 days after delivery compared to C.B-17 wild-type mouse livers 30 days after delivery.
 ii. C.B-17 wild-type mouse livers 30 days after delivery compared to C.B-17 SCID/beige mouse livers 30 days after delivery.

 - Two-way ANOVA comparing the percent of Nras-positive cells in C.B-17 wildtype and C.B-17 SCID/beige mouse livers 6 and 30 days after stable delivery of $Nras^{G12V/D38A}$.

2. Three-way ANOVA (2 × 2 × 2), comparing the percent of p21-positive cells in C.B-17 wild-type mouse livers vs C.B-17 SCID/beige mouse livers 6 and 30 days after stable delivery of $Nras^{G12V}$ or $Nras^{G12V/D38A}$.

 - Two-way ANOVA comparing the percent of p21-positive cells in C.B-17 wildtype and C.B-17 SCID/beige mouse livers 6 and 30 days after stable delivery of $Nras^{G12V}$.

 a. Planned comparisons with the Bonferroni correction:

 i. C.B-17 wild-type mouse livers 6 days after delivery compared to C.B-17 wild-type mouse livers 30 days after delivery.
 ii. C.B-17 wild-type mouse livers 30 days after delivery compared to C.B-17 SCID/beige mouse livers 30 days after delivery.

 - Two-way ANOVA comparing the percent of p21-positive cells in C.B-17 wildtype and C.B-17 SCID/beige mouse livers 6 and 30 days after stable delivery of $Nras^{G12V/D38A}$.

3. Three-way ANOVA (2 × 2 × 2), comparing the percent of p16-positive cells in C.B-17 wild-type mouse livers vs C.B-17 SCID/beige mouse livers 6 and 30 days after stable delivery of $Nras^{G12V}$ or $Nras^{G12V/D38A}$.

- Two-way ANOVA comparing the percent of p16-positive cells in C.B-17 wildtype and C.B-17 SCID/beige mouse livers 6 and 30 days after stable delivery of $Nras^{G12V}$.

   a. Planned comparisons with the Bonferroni correction:

      i. C.B-17 wild-type mouse livers 6 days after delivery compared to C.B-17 wild-type mouse livers 30 days after delivery.
      ii. C.B-17 wild-type mouse livers 30 days after delivery compared to C.B-17 SCID/beige mouse livers 30 days after delivery.

- Two-way ANOVA comparing the percent of p16-positive cells in C.B-17 wildtype and C.B-17 SCID/beige mouse livers 6 and 30 days after stable delivery of $Nras^{G12V/D38A}$.

## Known differences from the original study

The replication will only include evaluation of mice at days 6 and 30 post-injection. The original study also included evaluations at days 12 and 60 post-injection, as well as further analysis of liver tissue at 7 months post-injection. The replication is only comparing wild-type and SCID/beige mice; SCID mice were also included in the original study. The replication is including staining for Nras, p16, and p21. The original study also included pERK staining. All known differences in reagents are listed in the materials and reagents section above, as indicated in the comments section. All differences have the same capabilities as the original and are not expected to alter the experimental design.

## Provisions for quality control

Each of the transposon and transposase plasmids will be verified for sequence identity and DNA integrity. All immunohistochemistry experiments will include isotype antibody controls and secondary-only controls. All mice will be handled and housed in accordance with the Institutional Animal Care and Use Committee (IACUC) and the University of California-Davis. Exclusion criteria for successful hydrodynamic injections will be adhered to, as detailed above. According to the original authors, there is an expected failure rate of ~20% for hydrodynamic injection. Therefore, we are including extra animals in each treatment group so that we can maintain the samples sizes necessary to achieve high statistical power (see 'Power calculations'). All of the raw data, including immunohistochemistry controls and complete mouse health records, will be uploaded to the project page on the OSF (https://osf.io/82nfe/) and made publically available.

## Protocol 2: determining the significance of CD4+ T lymphocyte cells in senescence surveillance

This protocol assesses the specific necessity for CD4$^+$ T lymphocytes in immune surveillance of pre-malignant senescent hepatocytes. Oncogenic $Nras^{G12V}$ and non-oncogenic $Nras^{G12V/D38A}$ will be intrahepatically delivered to either wild-type or CD4$^{-/-}$ mice via hydrodynamic injection. Senescence surveillance will be determined by measuring the clearance of Nras-positive cells from liver tissue sections 12 days following injection. This protocol replicates experiments reported in Figure 4A.

## Sampling

1. These experiments will analyze a minimum of five mice per treatment group, for a total power of 94.6%.

- See 'Power calculations' section for details.
- In order to account for variability in hydrodynamic injections that might contribute to exclusion of study animals (estimated as a 20% failure rate by Kang et al.), the initial starting sample size will be seven mice per treatment group. The inclusion of two additional animals will help ensure that at least five animals are available for analysis at the conclusion of the study.

2.  Outline of experimental conditions: (tissue processed at 12 days post-injection).

    • 14 female C57/BL6 (H-2b) wild-type mice (4–6 weeks old).

       a. Seven mice injected with pPGK-SB13 + pT/CaggsNrasG12V.
       b. Seven mice injected with pPGK-SB13 + pT/CaggsNrasG12V/D38A.

    • 14 female C57/BL6 CD4$^{-/-}$ (4–6 weeks old).

       a. Seven mice injected with pPGK-SB13 + pT/CaggsNrasG12V.
       b. Seven mice injected with pPGK-SB13 + pT/CaggsNrasG12V/D38A.

## Materials and reagents

| Reagent | Type | Manufacturer | Catalog # | Comments |
| --- | --- | --- | --- | --- |
| pPGK-SB13 transposase vector | DNA Vector | N/A | N/A | Original reagent obtained from authors |
| pT/CaggsNrasG12V transposon vector | DNA Vector | N/A | N/A | Original reagent obtained from authors |
| pT/CaggsNrasG12V/D38A transposon vector | DNA Vector | N/A | N/A | Original reagent obtained from authors |
| Ketamine HCl | Anesthetic | | | Specific brand information will be left up to the discretion of the provider lab and recorded later |
| Acepromazine maleate | Anesthetic | | | |
| Butorphanol tartrate | Anesthetic | | | |
| Dulbecco's Phosphate Buffered Saline | Reagent | Sigma–Aldrich | D1408 | Original brand not specified |
| Paraformaldehyde | Reagent | Sigma–Aldrich | 158127 | Original brand not specified |
| C57/BL6J (H-2$^b$) | Mouse line | Jackson Laboratory | Strain #000664 | |
| C57/BL6 CD4$^{-/-}$ (H-2$^b$) (B6.129S2-CD4tm1Mak/J) | Mouse line | Jackson Laboratory | Strain #002663 | |
| Mouse anti-Nras (F155) | Antibody | Santa Cruz | Sc-31 | |
| Biotinylated goat anti-mouse IgG$_1$-conjugate | Antibody | Thermo-Fisher | TM-060-BN | |
| IgG1 mouse isotype control antibody | Antibody | Sigma–Aldrich | M5284 | |
| Diaminobenzidine (DAB) | IHC Stain | | | Specific brand information will be left up to the discretion of the provider lab and recorded later |
| Haematoxylin | IHC Stain | | | |
| Eosin | IHC Stain | | | |

## Procedure

1.  Obtain 4-week-old female C57/BL6 wild-type and C57/BL6 CD4$^{-/-}$ mice.

    a. House mice in individual ventilated cages (IVC) that are specific pathogen free (SPF). Allow mice 1 week to acclimate to new housing prior to injection.
    b. Mice should be randomized into strain-based treatment and control groups and co-housed so that mice of the same strain receiving either *Nras$^{G12V}$* or *Nras$^{G12V/D38A}$* injections are housed together.

2.  Using plasmids prepared in Protocol 1, randomly inject 5-week-old mice with transposon/transposase. Use detailed protocol provided in Protocol 1, as well as in the reference (***Bell et al., 2007***).

3.  At 12 days post-injection of DNA, harvest liver tissue from mice.

    a. Tissues should be randomized and blinded and shipped to Reveal Bioscience for downstream analysis.

4.  Fix, process, embed, and section tissues as described in Protocol 1.

5. Perform immunohistochemistry staining on each tissue section with the following:

   a. Primary antibody for N-ras.

      i. Mouse anti-Nras, 1:100 dilution.

   b. Secondary antibody.

      i. Biotinylated goat anti-mouse $IgG_1$-conjugate.

   c. Isotype control primary antibody + secondary antibody.

      i. IgG1 mouse isotype control antibody.

   d. Secondary antibody only.

6. Blindly examine five 200× fields from two stained liver sections from each mouse.

   a. Count 200 total cells per field and record number of Nras-positively stained cells for each image.

## Deliverables

- Data to be collected:

   1. Mouse records (weight at time of injection, duration(s) of injection, health over course of experiment, etc).
   2. Images of all stained sections, including controls.
   3. Raw total cell counts and Nras-positively stained cells for each field examined.
   4. Graph of Nras-positively stained cells as a percentage of total cells in field for each condition. (compare to Figure 4A).

## Confirmatory analysis plan

This replication attempt will perform the following statistical analyses listed below. First, the replication effect size will be computed and compared to the original effect size for each analysis. Second, the original effect size and the replication effect size will be combined into a single effect size using a meta-analytic approach and will be presented as a forest plot.

- Statistical analysis of the replication data:

   1. Two-way ANOVA, comparing the percent of Nras-positive cells in both wild-type and CD4$^{-/-}$ mice injected with Nras$^{G12V}$ vs wild-type and CD4$^{-/-}$ mice injected with Nras$^{G12V/D38A}$.

      - Planned comparisons with the Bonferroni correction.

         a. The percent of Nras-positive cells in wild-type mice injected with Nras$^{G12V}$ compared to the percent of Nras-positive cells in wild-type mice injected with Nras$^{G12V/D38A}$.
         b. The percent of Nras-positive cells in wild-type mice injected with Nras$^{G12V}$ compared to the percent of Nras-positive cells in CD4$^{-/-}$ mice injected with Nras$^{G12V}$.

   2. Note: In order to enable a direct comparison to the original data, an unpaired, two-tailed $t$-test (performed outside the framework of an ANOVA), comparing the percent of Nras-positive cells in wild-type mice injected with Nras$^{G12V}$ vs CD4$^{-/-}$ mice injected with Nras$^{G12V}$ will also be performed.

## Known differences from the original study

The replication will only include assessing genetically null CD4$^{-/-}$ mice and wild-type controls at a time point of 12 days following intrahepatic injections. The original study also evaluated α-CD4 antibody-depleted mice, as well as genetically null and antibody-depleted CD8$^{-/-}$ mice and genetically null Cd1d$^{-/-}$ mice. The original study also evaluated CD4$^{-/-}$ mice and wild-type mice 7 months after injection. All known differences in reagents are listed in the materials and reagents section above, as indicated in the comments section. All differences have the same capabilities as the original and are not expected to alter the experimental design.

## Provisions for quality control

Each of the transposon and transposase plasmids will be verified for sequence identity and DNA integrity. All immunohistochemistry experiments will include isotype antibody controls and secondary-only controls. All mice will be handled and housed in accordance with the Institutional Animal Care and Use Committee (IACUC) and the University of California-Davis. Exclusion criteria for successful hydrodynamic injections will be adhered to, as detailed above. According to the original authors, there is an expected failure rate of ~20% for hydrodynamic injection. Therefore, we are including extra animals in each treatment group so that we can maintain the samples sizes necessary to achieve high statistical power (see 'Power calcualtions'). All of the raw data, including immunohistochemistry controls and complete mouse health records will be uploaded to the project page on the OSF (https://osf.io/82nfe/) and made publically available.

## Power calculations

### Protocol 1

Summary of original data from Figure 3B (**Kang et al., 2011**):

| Nras-positive hepatocytes (Figure 3B) | Mean | SD | N |
|---|---|---|---|
| CB17 wild-type injected with NrasG12V (Day 6) | 15.4 | 3.9 | 4 |
| CB17 wild-type injected with NrasG12V (Day 30) | 1.2 | 0.7 | 4 |
| SCID/beige injected with NrasG12V (Day 6) | 16.4 | 2.8 | 4 |
| SCID/beige injected with NrasG12V (Day 30) | 12.8 | 2.9 | 4 |

### Test family

2-way ANOVA: Fixed effects, special, main effects and interactions, alpha error = 0.05.

- Power calculations were performed with effects reported in original study using G*Power software (version 3.1.7) (**Faul et al., 2007**).
- ANOVA F statistic calculated with GraphPad Prism 6.0.
- Partial $\eta^2$ calculated from **Lakens (2013)**.

### Power calculations for replication

| Group | F (DFn, Dfd) | Partial $\eta^2$ | Effect size f | A priori power | Total Sample Size |
|---|---|---|---|---|---|
| NrasG12V | F(1, 12) = 14.0670 (interaction) | 0.539648 | 1.082705 | 90.6%* | 12* (3/group) |

*20 total (5/group) will be used based on the p21 planned comparison calculations making the power 99.5%.

### Test family

- 2 tailed *t* test, difference between two independent means, Bonferroni's correction: alpha error = 0.025.

### Power calculations (performed with G*Power software, version 3.1.7 [**Faul et al., 2007**]).

| Group 1 | Group 2 | Effect size d | A priori power | Group 1 sample size | Group 2 sample size |
|---|---|---|---|---|---|
| CB17/G12V/Day 6 | CB17/G12V/Day 30 | 5.068197 | 96.5%* | 3* | 3* |
| CB17/G12V/Day 6 | SCIDBeige/G12V/Day 30 | 5.498927 | 98.3%* | 3* | 3* |

*Five in each group will be used based on the p21 planned comparison calculations making the power 99.9%.

## Summary of original data from Figure 3C (*Kang et al., 2011*):

| p21-positive hepatocytes (Figure 3C) | Mean | SD | N |
|---|---|---|---|
| CB17 wild-type injected with NrasG12V (Day 6) | 15.5 | 2.0 | 4 |
| CB17 wild-type injected with NrasG12V (Day 30) | 0.8 | 1.7 | 4 |
| SCID/beige injected with NrasG12V (Day 6) | 15.3 | 1.9 | 4 |
| SCID/beige injected with NrasG12V (Day 30) | 8.8 | 3.9 | 4 |

### Test family

2-way ANOVA: Fixed effects, special, main effects and interactions, alpha error = 0.05.

- Power calculations were performed with effects reported in original study using G*Power software (version 3.1.7) (*Faul et al., 2007*).
- ANOVA F statistic calculated with GraphPad Prism 6.0.
- Partial $\eta^2$ calculated from *Lakens (2013)*.

### Power calculations for replication

| Group | F (DFn, Dfd) | Partial $\eta^2$ | Effect size f | A priori power | Total Sample Size |
|---|---|---|---|---|---|
| NrasG12V | F(1, 12) = 10.4613 (interaction) | 0.465748 | 0.9336894 | 80.8%* | 12* (3/group) |

*20 total (5/group) will be used based on the planned comparison calculations making the power 97.5%.

### Test family

- 2 tailed *t* test, difference between two independent means, Bonferroni's correction: alpha error = 0.025.

### Power calculations (performed with G*Power software, version 3.1.7 [*Faul et al., 2007*]).

| Group 1 | Group 2 | Effect size d | A priori power | Group 1 sample size | Group 2 sample size |
|---|---|---|---|---|---|
| CB17/G12V/Day 6 | CB17/G12V/Day 30 | 7.919955 | 99.9%* | 3* | 3* |
| CB17/G12V/Day 6 | SCIDBeige/G12V/Day 30 | 2.659290 | 89.8% | 5 | 5 |

*Five in each group will be used based on the planned comparison calculations making the power 99.9%.

## Summary of original data from Figure 3E (*Kang et al., 2011*):

| p16-positive hepatocytes (Figure 3E) | Mean | SD | N |
|---|---|---|---|
| CB17 wild-type injected with NrasG12V (Day 6) | 14.5 | 1.9 | 4 |
| CB17 wild-type injected with NrasG12V (Day 30) | 0 | 0 | 4 |
| SCID/beige injected with NrasG12V (Day 6) | 15.8 | 2.0 | 4 |
| SCID/beige injected with NrasG12V (Day 30) | 10.6 | 4.0 | 4 |

### Test family

2-way ANOVA: Fixed effects, special, main effects and interactions, alpha error = 0.05.

- Power calculations were performed with effects reported in original study using G*Power software (version 3.1.7) (*Faul et al., 2007*).

- ANOVA F statistic calculated with GraphPad Prism 6.0.
- Partial $\eta^2$ calculated from *Lakens (2013)*.

## Power calculations for replication

| Group | F (DFn, Dfd) | Partial $\eta^2$ | Effect size f | A priori power | Total Sample Size |
|---|---|---|---|---|---|
| NrasG12V | F(1, 12) = 14.6531 (interaction) | 0.549771 | 1.105030 | 91.6%* | 12* (3/group) |

*20 total (5/group) will be used based on the p21 planned comparison calculations making the power 99.6%.

## Test family

- 2 tailed *t* test, difference between two independent means, Bonferroni's correction: alpha error = 0.025.

## Power calculations (performed with G*Power software, version 3.1.7 [*Faul et al., 2007*]).

| Group 1 | Group 2 | Effect size d | A priori power | Group 1 sample size | Group 2 sample size |
|---|---|---|---|---|---|
| CB17/G12V/Day 6 | CB17/G12V/Day 30 | 10.79268 | 99.9%* | 3* | 3* |
| CB17/G12V/Day 6 | SCIDBeige/G12V/Day 30 | 3.747666 | 80.1%† | 3† | 3† |

*Five in each group will be used based on the p21 planned comparison calculations making the power 99.9%.
†Five in each group will be used based on the p21 planned comparison calculations making the power 99.6%.

## Protocol 2
Summary of original data from Figure 4A (*Kang et al., 2011*):

| Nras-positive hepatocytes (Figure 4A) | Mean | SD | N |
|---|---|---|---|
| BL/6 wild-type injected with *Nras*$^{G12V}$ | 6.1 | 1.7 | 5 |
| BL/6 wild-type injected with *Nras*$^{G12V/D38A}$ | 16.2 | 2.3 | 5 |
| CD4$^{-/-}$ injected with *Nras*$^{G12V}$ | 18.6 | 3.7 | 5 |
| CD4$^{-/-}$ injected with *Nras*$^{G12V/D38A}$ | 18.4 | 4.9 | 5 |

## Test family
2-way ANOVA: Fixed effects, special, main effects, and interactions, alpha error = 0.05.

- Power calculations were performed with effects reported in original study using G*Power software (version 3.1.7) (*Faul et al., 2007*).
- ANOVA F statistic calculated with GraphPad Prism 6.0.
- Partial $\eta^2$ calculated from *Lakens (2013)*.

## Power calculations for replication

| F (DFn, Dfd) | Partial $\eta^2$ | Effect size f | A priori power | Total sample size |
|---|---|---|---|---|
| F(1, 16) = 11.5617 (interaction) | 0.419484 | 0.850062 | 87.7%[1] | 16* (4/group) |

*20 total (5/group) will be used to account for additional variance making the power 94.6%.

Test family

- 2 tailed *t* test, difference between two independent means, Bonferroni's correction: alpha error = 0.025.

Power calculations (Performed with G*Power software, version 3.1.7 [*Faul et al., 2007*]).

| Group 1 | Group 2 | Effect size d | A priori power | Group 1 sample size | Group 2 sample size |
|---|---|---|---|---|---|
| BL/6 G12V | BL/6 G12V/D38A | 4.994129 | 96.0%* | 3* | 3* |
| BL/6 G12V | CD4$^{-/-}$ G12V | 4.341429 | 90.0%* | 3* | 3* |

*Five in each group will be used to account for additional variance making the power 99.9%.

# Acknowledgements

The Reproducibility Project: Cancer Biology core team would like to thank the original authors, in particular Lars Zender and Tae-Won Kang, for generously sharing critical information and reagents to ensure the fidelity and quality of this replication attempt. We are grateful to Courtney Soderberg at the Center for Open Science for assistance with statistical analyses. We would also like to thank the following companies for generously donating reagents to the Reproducibility Project: Cancer Biology; American Type Culture Collection (ATCC), BioLegend, Cell Signaling Technology, Charles River Laboratories, Corning Incorporated, DDC Medical, EMD Millipore, Harlan Laboratories, LI-COR Biosciences, Mirus Bio, Novus Biologicals, Sigma–Aldrich, and System Biosciences (SBI).

# Additional information

### Group author details

**Reproducibility Project: Cancer Biology**

Elizabeth Iorns: Science Exchange, Palo Alto, California; William Gunn: Mendeley, London, United Kingdom; Fraser Tan: Science Exchange, Palo Alto, California; Joelle Lomax: Science Exchange, Palo Alto, California; Timothy Errington: Center for Open Science, Charlottesville, Virginia

### Competing interests
SR: The UC-Davis Mouse Biology Program is a Science Exchange affiliated lab. CW: Reveal Biosciences is a Science Exchange affiliated lab. RP:CB: EI, FT and JL are employed and holds shares in Science Exchange, Inc. The other authors declare that no competing interests exist.

### Funding

| Funder | Author |
|---|---|
| Laura and John Arnold Foundation | Reproducibility Project: Cancer Biology |

The Reproducibility Project: Cancer Biology is funded by the Laura and John Arnold Foundation, provided to the Center for Open Science in collaboration with Science Exchange. The funder had no role in study design or the decision to submit the work for publication.

### Author contributions
SR, CW, NY, Drafting or revising the article; RP:CB, Conception and design, Drafting or revising the article

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
