## [Decision Letter]

Thank you for sending your work entitled “Registered report: Senescence
surveillance of pre-malignant hepatocytes limits liver cancer development” for
consideration at *eLife*. Your article has been favorably evaluated by
Sean Morrison (Senior editor), a Reviewing editor, and 4 reviewers, one of whom is a
statistician.

The Reviewing editor and the reviewers discussed their comments before we reached this
decision, and the Reviewing editor has assembled the following comments to help you
prepare a revised submission. All believe this is a generally very well-conceived plan
for replication, but also raised some specific issues that need to be addressed before a
protocol is finalized, and the registered report accepted for publication in
*eLife*.

Major issues to be addressed in a revised submission:

Technical points:

1) As the proposed reproduction experiments involve hydrodynamic tail vein injection, it
is crucial that these experiments are conducted in a laboratory with high experience
with these technique. It is important to note that the efficiency of DNA uptake is
influenced by blood pressure and heart rate. Preferably, the technique is conducted
without anesthesia, as in this way there will be no drop in blood pressure, and the
cardiovascular system of the mouse can fully adapt to the volume challenge by
hydrodynamic tail vein injection. In any case, two studies can only be compared if the
same anesthesia is used. The anesthesia proposed for the reproduction experiment
involves, in addition to ketamine, acepromazine. As acepromazine is known to induce
vasodilation and decreases blood pressure, it cannot be ruled out that acepromazine
impacts the efficiency of hydrodynamic DNA delivery. This is important to keep in mind,
as in the original paper by Kang et al. I could not find information that acepromazine
was used.

2) To ensure comparability of two studies it is furthermore important that mice of the
same age and weight are subjected to hydrodynamic delivery. Kang et al. describe that
mice 4–6 weeks of age were used for hydrodynamic delivery while in the outline of
the reproducibility project it is stated that mice 4–6 weeks or 6–8 weeks
of age are ordered and used after one week of adjustment time. It should be ensured that
mice are injected at the same age as described by Kang et al. Likewise, the hydrodynamic
injections are a critical step in this experiment. The researcher who conducts these
injections should learn this method in a laboratory in which this protocol is
well-established, if possible the Zender lab itself.

3) Even though the mice are being caged under SPF conditions, there is no evidence that
the research team will test for the presence of viruses that are largely confined to
mice with highly significant immunodeficiencies. These can often have significant
effects on the anti-tumor activity of innate immunity. At a minimum that the
investigators should test for the presence of Norovirus in their SCID/Beige mice as a
marker for such infections, especially since Norovirus is found in many SPF colonies
across the country. The experimental groups should also be co-caged to control, at least
partially, for strain specific microbiota that can also have profound influences on
natural anti-tumor activity.

4) With respect to Protocol 2, Kang et al. went to extreme lengths to carefully look at
the role of class II restricted CD4 T cells, including using
CD4^−/−^ mice, Class II^–/–^ mice, and
wt mice depleted of CD4^+^ cells using CD4 specific mAb. All of their
results agree that CD4^+^ T cells are playing a major role in
surveillance. However, in the proposed study, the research team only proposes to use
CD4^−/−^ mice. There may be effectors or Tregs that emerge
from these mice that can still influence the response, and at least one more of the
previously employed models of mice lacking CD4 T cells should also be used to
substantiate the effects that may or may not be seen with
CD4^−/−^ mice.

Statistical issues:

1) The authors state that: “The original authors of the study performed this
[statistical] analysis, and we are therefore including it in the replication analysis
plan. We are also including the more statistically appropriate tests detailed
below”. The used wording implies that the authors of the replication study regard
the statistical analyses by Kang et al. as not adequate. More detailed information
should be provided as to why the authors think the analyses by Kang et al. are not
adequate.

2) Each protocol has statistics mentioned in the confirmatory analysis plan and the
power calculation sections. In protocol 1 the analyses will be two-way ANOVA applied to
three datasets. In each ANOVA are the effects treatment and time? The summary data from
the original report suggests this. Means and SD are reported for each of the four groups
within each experiment but the factor specific effects are not reported. A two-way ANOVA
can consist of up to three tests, the treatment, the time, and the treatment–time
interaction. The power calculations use a single effect size, which we do not
understand. As the experiment is trying to replicate an already detected effect, then
each such effect can be clearly stated. The power calculation is reported for one degree
of freedom suggesting only one of these terms (treatment?) is expected to be significant
though time clearly seems to have a strong effect in one group.

3) The plans for protocol 2 are easier to understand in that simple one-way ANOVAs and
t-tests are planned, though with just 5 per group this will rely on the very strong
assumption of normality. Prior experience involving tumor in vivo studies shows that
such experiments have considerable mouse-to-mouse variation. It would be a shame if
conclusions are limited by too small sample numbers, and in light of the statistical
issue just raised, it seems important to increase the animal numbers per group to n
= 10–15 because of the low projecting 81% confidence level at n = 5.
Also, the power calculations seem to justify 3 in each sample, yet 5 in each group are
planned. The authors need to better explain their power calculations that led them to
conclude openly 3 animals per group would be sufficient and why they then choose 5
animals per group. Is this to cover some results lost due to failure? In any case, the
power calculations for these comparisons need to state what specific effect sizes will
be viewed as a replication of the original study.

4) In a related vein, in a replication study it is important to be clear what specific
cut offs will be used to declare a replication or lack of replication. So the explicit
effect sizes for each factor and for each of the 6 experiments need to be stated. While
it is a good idea to combine the two results (original and replication) in a forest
plot, this will give an inflated estimate of any effect sizes due to the publication
selection operating on the first article. The decision regarding whether there has been
replication should be based on the results of the replication study alone. So it may be
that smaller effect sizes than those reported in the original article would still be
viewed as strong evidence of the anticipated effect. In such a case, as stated above
(point #3), a larger sample size would be justified and should be strongly
considered at this stage.

5) Regarding the power calculations, the proposed power calculations use a combination
of data reported in the original study and data from an alternative relevant study. This
is fine at this stage, however, we suggest the following improvements:

a) Cross-study variation should be taken into account to determine expected power in the
proposed study, since loss of power can be expected from the original study. This is
hard to estimate, but papers by Giovanni Parmigiani and collaborators at the
Dana–Farber provide some estimates about cross-study variation that could be used
for this purpose. The authors should budget some additional variability because of
cross-study reproducibility, and increase the sample size on-the-fly, as they deem
appropriate.

b) The final report on the replicated study should report the actual power of the tests,
based on numerical summaries of the data in the replicated study.

[Editors' note: further revisions were requested prior to acceptance, as described
below.]

Thank you for resubmitting your article entitled “Registered report: Senescence
surveillance of pre-malignant hepatocytes limits liver cancer development” for
further consideration at *eLife*. Your revised Registered report has been
favorably evaluated by Sean Morrison (Senior editor), a member of the Board of Reviewing
Editors, and by someone with relevant expertise in statistics.

The Registered report is essentially ready for acceptance, but we would ask you to
respond to the remaining statistical issues and make additional minor modifications as
needed:

1) We do not think it is a good idea to report a combined estimate of any effect size
with the original. However displaying results on a forest plot is helpful.

2) With respect to the sample size calculations for the two-way ANOVA, have you used the
within group SDs instead of the within group variances in the formulae? If so, this
means that you would be over-estimating the power. With 5 in each group the authors
should just have 80% power to replicate what was seen in the original study.

---

## [Author Response]

*1) As the proposed reproduction experiments involve hydrodynamic tail vein
injection, it is crucial that these experiments are conducted in a laboratory with
high experience with these technique. It is important to note that the efficiency of
DNA uptake is influenced by blood pressure and heart rate. Preferably, the technique
is conducted without anesthesia, as in this way there will be no drop in blood
pressure, and the cardiovascular system of the mouse* can *fully adapt
to the volume challenge by hydrodynamic tail vein injection. In any case, two
studies* can *only be compared if the same anesthesia is used. The
anesthesia proposed for the reproduction experiment involves, in addition to
ketamine, acepromazine. As acepromazine is known to induce vasodilation and decreases
blood pressure, it cannot be ruled out that acepromazine impacts the efficiency of
hydrodynamic DNA delivery. This is important to keep in mind, as in the original
paper by Kang et al. I could not find information that acepromazine was
used*.

We thank the reviewers for this suggestion. According to correspondence with the
original authors, Kang and colleagues followed the protocol of [2] in order to perform their hydrodynamic
injections. This protocol clearly outlines a drug cocktail of 8 mg
ml^−1^ ketamine HCl, 0.1 mg ml^−1^ acepromazine
maleate and 0.01 mg ml^−1^ butorphanol tartrate. We agree with the
reviewers that deviating from the exact drug cocktail used in the original study may
contribute to skewed results; therefore, we will also follow the protocol of Bell, et
al., and use the same anesthetics as used in the original study.

We have included a biographical sketch of the scientists directly in charge of
performing the hydrodynamic injections. Ms. Lynette Bower has over ten years of
experience in laboratory animal procedures, and is certified proficient in performing
HPTV injections. She has performed over 50 injections on four separate projects, with a
>80% success rate. Ms. Bower will be overseen by Dr. Kristin Evans, who is one of
the directors of the Mouse Biology Program at the University of California, Davis.

In order to be overly conservative in our approach, we have increased the sample sizes
for each cohort of mice receiving hydrodynamic injections from 5 animals to 7. In this
way, we feel confident that we will have enough end-point animals to achieve high
statistical power in our analyses.

*2) To ensure comparability of two studies it is furthermore important that mice
of the same age and weight are subjected to hydrodynamic delivery. Kang et al.
describe that mice 4–6 weeks of age were used for hydrodynamic delivery while
in the outline of the reproducibility project it is stated that mice 4–6 weeks
or 6–8 weeks of age are ordered and used after one week of adjustment time. It
should be ensured that mice are injected at the same age as described by Kang et al.
Likewise, the hydrodynamic injections are a critical step in this experiment. The
researcher who conducts these injections should learn this method in a laboratory in
which this protocol is well-established, if possible the Zender lab
itself*.

We thank the reviewers for calling our attention to this apparent discrepancy. While our
initial correspondence with Kang and colleagues implied that two age ranges of mice were
used, subsequent communication with Dr. Tae Won Kang has revealed that for the
particular experiments outlined in this replication, all mice should be 4–6 weeks
of age.

We have clarified the exact ages of mice that will be used for each protocol in the
Registered report. For both Protocol 1 and 2, female mice will be ordered at 4 weeks of
age, acclimated for 1 week, and injected at 5 weeks of age. The precise ages and weights
of mice at the time of injection will be documented.

In regards to performing hydrodynamic injections, we will follow the procedures outlined
in [2], as followed by Kang and
colleagues. The replicating scientists are highly experienced in the practice of
hydrodynamic injections (please see above response). Additionally, we are increasing the
sample sizes for all animals receiving injections, so as to ensure the statistical power
of our downstream analyses (see above response).

*3) Even though the mice are being caged under SPF conditions, there is no
evidence that the research team will test for the presence of viruses that are
largely confined to mice with highly significant immunodeficiencies. These*
can *often have significant effects on the anti-tumor activity of innate
immunity. At a minimum that the investigators should test for the presence of
Norovirus in their SCID/Beige mice as a marker for such infections, especially since
Norovirus is found in many SPF colonies across the country. The experimental groups
should also be co-caged to control, at least partially, for strain specific
microbiota that* can *also have profound influences on natural
anti-tumor activity*.

We thank the reviewers for these suggestions. We have attached a sample health report
from the University of California–Davis Mouse Biology Program (UCD-MBP) that
details the testing parameters in their vivarium. As noted, murine norovirus (MNV) is
routinely tested for via serum ELISA. UCD-MBP has not detected any pathogens in their
colonies.

We agree with the reviewers’ suggestion to co-cage experimental and control
groups of mice within the same strain. Therefore, we have updated the Registered report
to indicate that mice of the same strain receiving either
*Nras*^*G12V*^ or
*Nras*^*G12V/D38A*^ will be co-caged both
before and after receiving injections. As far as co-caging mice between strains, we
believe this introduces a major procedural difference that did not occur in the original
study, as strain mixing may introduce variation in mouse behavior and/or microbiota (as
noted by the reviewer). To minimize such differences in our replication, we will
continue to house mice strain-specifically.

*4) With respect to Protocol 2, Kang et al. went to extreme lengths to carefully
look at the role of class II restricted CD4 T cells, including using
CD4*^−/−^
*mice, Class II*^–/–^
*mice, and wt mice depleted of CD4*^*+*^
*cells using CD4 specific mAb. All of their results agree that
CD4*^*+*^
*T cells are playing a major role in surveillance. However, in the proposed
study, the research team only proposes to use
CD4*^−/−^
*mice. There may be effectors or Tregs that emerge from these mice that*
can *still influence the response, and at least one more of the previously
employed models of mice lacking CD4 T cells should also be used to substantiate the
effects that may or may not be seen with CD4*^−/−^
*mice*.

We thank the reviewers for this insightful comment, and we agree that this is an
important issue to consider during data interpretation. Although we plan to obtain mice
from the same vendor as the original authors (Jackson Laboratory), there may be subtle
differences that emerge even from seemingly genetically identical animals. The raised
concern may be a reason why the experimental effect, or size of the effect, is not
replicated and thus will be acknowledged as a potential factor in the Replication Study.
Alternatively, the raised concern may result in the same observed effect, but not due to
the role of CD4 cells. This too will be acknowledged in the Replication Study.

We recognize that all of the experiments included in the original study are important,
and choosing which experiments to replicate has been one of the great challenges of this
project. We agree that the exclusion of certain experiments limits the scope of what can
be analyzed by the project, but we are attempting to identify a balance of breadth of
sampling for general inference with sensible investment of resources on replication
projects. The Reproducibility Project: *Cancer* Biology is aimed to
replicate a selection of experiments as faithfully as possible, not necessarily the main
conclusions that are drawn from the many experiments in any given paper.

*Statistical issues*:

*1) The authors state that: “The original authors of the study performed
this [statistical] analysis, and we are therefore including it in the replication
analysis plan. We are also including the more statistically appropriate tests
detailed below”. The used wording implies that the authors of the replication
study regard the statistical analyses by Kang et al. as not adequate. More detailed
information should be provided as to why the authors think the analyses by Kang et
al. are not adequate*.

We have updated the language in the Registered report to better reflect the strategy for
this analysis. As outlined in Nieuwenhuis et al. (Nieuwenhuis et al., 2011), the authors
did not report an interaction effect (Nras mutational status–immune defect),
which the researchers would have needed to report as significant to support their claim.
Thus, we will be performing the two-way ANOVA to determine if the interaction is
significant before performing pair-wise comparisons. However, we also plan to perform a
t-test outside the framework of an ANOVA similar to the original paper (although
performing a planned comparison within the framework of an ANOVA is more powerful than
performing a separate t-test if the assumption of ANOVA is valid) to allow for a direct
comparison to what was originally reported.

*2) Each protocol has statistics mentioned in the confirmatory analysis plan and
the power calculation sections. In protocol 1 the analyses will be two-way ANOVA
applied to three datasets. In each ANOVA are the effects treatment and time? The
summary data from the original report suggests this. Means and SD are reported for
each of the four groups within each experiment but the factor specific effects are
not reported. A two-way ANOVA* can *consist of up to three tests, the
treatment, the time, and the treatment-time interaction. The power calculations use a
single effect size, which we do not understand. As the experiment is trying to
replicate an already detected effect, then each such effect* can *be
clearly stated. The power calculation is reported for one degree of freedom
suggesting only one of these terms (treatment?) is expected to be significant though
time clearly seems to have a strong effect in one group*.

We thank the reviewer for these helpful comments. We have revised our confirmatory
analysis plan for Protocol 1, as updated in the Registered report. We have also updated
the Power Calculations section to better reflect both the analysis we performed on the
original data, as well as our future analysis plans. The original data had a significant
treatment–time interaction, which we are powered for and that we have more
clearly labeled. Thus, the experiment is designed to test the originally observed effect
of time differing between the treatments. However, we do see value in adding planned
comparisons between the two times for each treatment as this will evaluate the size of
the effect in each group. Therefore, we have included such comparisons in the updated
version of our confirmatory analysis plan for Protocol 1.

*3) The plans for protocol 2 are easier to understand in that simple one-way
ANOVAs and t-tests are planned, though with just 5 per group this will rely on the
very strong assumption of normality. Prior experience involving tumor* in
vivo *studies shows that such experiments have considerable mouse-to-mouse
variation. It would be a shame if conclusions are limited by too small sample
numbers, and in light of the statistical issue just raised, it seems important to
increase the animal numbers per group to n= 10-15 because of the low projecting
81% confidence level at n = 5. Also, the power calculations seem to justify 3 in
each sample, yet 5 in each group are planned. The authors need to better explain
their power calculations that led them to conclude openly 3 animals per group would
be sufficient and why they then choose 5 animals per group. Is this to cover some
results lost due to failure? In any case, the power calculations for these
comparisons need to state what specific effect sizes will be viewed as a replication
of the original study*.

We thank the reviewers for these insightful suggestions. We agree that there is the
potential for unanticipated variation in the experimental design that would lead to the
exclusion of animals in a given group. Regarding hydrodynamic injections, the original
study authors communicated variation in injections that resulted in a ∼20%
injection failure rate, leading to exclusion of animals in downstream analyses. To
account for variation in injections, we have increased the sample size to seven mice per
treatment group for the experiments performed in Protocol 1, as well as the 12-day
experimental arm of Protocol 2. We believe this increased sample size will allow us to
analyze a minimum of five mice per group in downstream analyses. We have updated the
Power Calculations to indicate the statistical power achieved by including five mice per
group (89.9–99.9% for Protocol 1; 94.6–99.9% for Protocol 2).

We believe the reviewers have highlighted an important concern regarding the potential
for mouse-to-mouse variation in tumor development in the 7-month experimental arm of
Protocol 2. Given that we cannot calculate the effect size of the original experimental
data in Figure 4B (specifically the CD4^−/−^ arm), and that we
also do not know the variance or the distribution of the data for this arm, it is
difficult to properly power this replication with a large enough sample size that also
remains within the feasibility of the project. Based on these factors, we have
reconsidered the inclusion of the 7-month in vivo tumor formation experiment in this
replication study. Given that the original effect size is unknown, we feel that a
replication of this experiment would be considered exploratory in nature; thus, we have
reallocated our resources toward increasing sample sizes for, and therefore improving,
those arms of the study that can be statistically compared to the original data.

*4) In a related vein, in a replication study it is important to be clear what
specific cut offs will be used to declare a replication or lack of replication. So
the explicit effect sizes for each factor and for each of the 6 experiments need to
be stated. While it is a good idea to combine the two results (original and
replication) in a forest plot, this will give an inflated estimate of any effect
sizes due to the publication selection operating on the first article. The decision
regarding whether there has been replication should be based on the results of the
replication study alone. So it may be that smaller effect sizes than those reported
in the original article would still be viewed as strong evidence of the anticipated
effect. In such a case, as stated above (point #3), a larger sample size would
be justified and should be strongly considered at this stage*.

We agree that the replication effect size and the original effect size (with their
corresponding 95% confidence intervals) should be compared to each other. Therefore,
each confirmatory analysis plan section begins with the phrasing describing the
comparison of the two effect sizes and the combination of them. We have slightly
modified the text in the Registered report to clarify this plan. We plan to present the
effect sizes of each factor for all the experiments this way and to combine each factor
(original and replication) using a meta-analytic approach. This will allow direct
comparison of the original and replication effect sizes and the combination of the two
will give an estimation of the current knowledge for that given effect size. And we
agree that the direct comparison will allow for one to determine if the evidence for an
anticipated effect is still there, even if the size of that effect, or precision of that
effect size estimate is different.

*5) Regarding the power calculations, the proposed power calculations use a
combination of data reported in the original study and data from an alternative
relevant study. This is fine at this stage, however, we suggest the following
improvements*:

*a) Cross-study variation should be taken into account to determine expected
power in the proposed study, since loss of power* can *be expected
from the original study. This is hard to estimate, but papers by Giovanni Parmigiani
and collaborators at the Dana–Farber provide some estimates about cross-study
variation that could be used for this purpose. The authors should budget some
additional variability because of cross-study reproducibility, and increase the
sample size on-the-fly, as they deem appropriate*.

We thank the reviewers for these suggestions. The cross-study variation, such as
approaches that utilize the 95% confidence interval of the effect size, can be useful in
conducting power calculations when planning adequate sample sizes for detecting the true
population effect size, which requires a range of possible observed effect sizes.
However, the Reproducibility Project: *Cancer* Biology is designed to
conduct replications that have 80% power to detect the point estimate of the originally
reported effect size. While this has the limitation of being underpowered to detect
smaller effects than what is originally reported, this standardizes the approach across
all studies to be designed to detect the originally reported effect size with at least
80% power. Also, while the minimum power guarantee is beneficial for observing a range
of possible effect sizes, the experiments in this replication, and all experiments in
the project, are designed to detect the originally reported effect size with a minimum
power of 80%. Thus, performing power calculations during or after data collection is not
necessary in this replication attempt as all studies included are already designed to
meet a minimum power or are identified beforehand as being underpowered and thus are not
included in the confirmatory analysis plan. The papers by Giovanni Parmigiani and
collaborators highlight the importance of accounting for variability that can occur
across different studies, specifically gene expression data. While it is possible for a
difference in variance between the originally reported results and the replication data,
this will be reflected in the presentation of the data and a possible reason for
obtaining a different effect size estimate.

*b) The final report on the replicated study should report the actual power of
the tests, based on numerical summaries of the data in the replicated
study*.

As described above, we do not see the value in performing post-hoc power calculations on
the obtained data. However, we do agree that reporting the actual power of the tests to
detect the originally reported effect size estimate based on the sample size analyzed in
the replication study is important and will be reported.

[Editors’ note: further revisions were requested prior to acceptance, as
described below.]

*1) We do not think it is a good idea to report a combined estimate of any effect
size with the original. However displaying results on a forest plot is
helpful*.

We disagree, and as described in Valentine et al., 2011, and Bumming, 2012, combining
multiple studies using a meta-analytic approach is a statistical option that can be
employed to describe all of the available evidence about a given effect size.
Specifically, we will utilize a random effects meta-analysis because this approach
assumes that the effects vary due to known and unknown characteristics of the
studies.

*2) With respect to the sample size calculations for the two-way ANOVA, have you
used the within group SDs instead of the within group variances in the formulae? If
so, this means that you would be over-estimating the power. With 5 in each group the
authors should just have 80% power to replicate what was seen in the original
study*.

We used the SD when we calculated the F statistic, which is the input for GraphPad
Prism’s software when the original data values are not known. We also obtained
the same F values and partial eta squared values when using the R software package
‘rpsychi’ and the function ‘ind.twoway.second’, which
conducts a two-way design using published work using the mean, SD, and n (http://cran.r-project.org/package=rpsychi). Please see the
“Study 34 2-way ANOVAs.R” file.

*References*:

Valentine JC, Biglan A, Boruch RF, Castro FG, Collins LM, Flay BR, Kellam S, Moscicki
EK, Schinke SP. 2011. Replication in prevention science. Prev Sci. 12(2): 103-17.
doi:10.1007/s11121-011-0217-6.

Bumming G. 2012. Understanding The New Statistics: Effect Sizes, Confidence Intervals,
and Meta-Analysis. Routledge. ISBN: 978-0-415-87,968-2.